# DECISION BOUNDARY ANALYSIS OF ADVERSARIAL EXAMPLES

**Warren He, Bo Li & Dawn Song**
Computer Science Division
University of California, Berkeley

## ABSTRACT

Deep neural networks (DNNs) are vulnerable to *adversarial examples*, which are carefully crafted instances aiming to cause prediction errors for DNNs. Recent research on adversarial examples has examined local neighborhoods in the input space of DNN models. However, previous work has limited what regions to consider, focusing either on low-dimensional subspaces or small balls. In this paper, we argue that information from larger neighborhoods, such as from more directions and from greater distances, will better characterize the relationship between adversarial examples and the DNN models. First, we introduce an attack, OPT-MARGIN, which generates adversarial examples robust to small perturbations. These examples successfully evade a defense that only considers a small ball around an input instance. Second, we analyze a larger neighborhood around input instances by looking at properties of surrounding decision boundaries, namely the distances to the boundaries and the adjacent classes. We find that the boundaries around these adversarial examples do not resemble the boundaries around benign examples. Finally, we show that, under scrutiny of the surrounding decision boundaries, our OPTMARGIN examples do not convincingly mimic benign examples. Although our experiments are limited to a few specific attacks, we hope these findings will motivate new, more evasive attacks and ultimately, effective defenses.

## 1 INTRODUCTION

Recent research in adversarial examples in deep learning has examined local neighborhoods in the input space of deep learning models. Liu et al. (2017) and Tramèr et al. (2017) examine limited regions around benign samples to study why some adversarial examples transfer across different models. Madry et al. (2017) explore regions around benign samples to validate the robustness of an adversarially trained model. Tabacof & Valle (2016) examine regions around adversarial examples to estimate the examples' robustness to random noise. Cao & Gong (2017) determine that considering the region around an input instance produces more robust classification than looking at the input instance alone as a single point.

These previous works have limited what regions they consider. Liu et al. and Tramèr et al. focus on low-dimensional subspaces around a model's gradient direction. Tabacof & Valle and Cao & Gong explore many directions, but they focus on a small ball.

In this paper, we argue that information from larger neighborhoods—both in more directions and at greater distances—will better help us understand adversarial examples in high-dimensional datasets.

First, we describe a concrete limitation in a system that utilizes information in small neighborhoods. Cao & Gong's region classification defense (2017) takes the majority prediction in a small ball around an input instance. We introduce an attack method, OPTMARGIN, for generating adversarial examples that are robust to small perturbations, which can evade this defense.

Second, we provide an example of how to analyze an input instance's surroundings in the model's input space. We introduce a technique that looks at the decision boundaries around an input instance, and we use this technique to characterize our robust OPTMARGIN adversarial examples. Our analysis reveals that, while OPTMARGIN adversarial examples are robust enough to fool region

classification, the decision boundaries around them do not resemble the boundaries around benign examples, in terms of distances from the example to the adjacent classes.

Third, as an extension to the above observation, we train a classifier to differentiate the decision boundary information that comes from different types of input instances. We show that our classifier can differentiate OPTMARGIN and benign examples with 90.4% accuracy, whereas region classification limits itself to a small region and fails. However, it remains to be seen whether a more sophisticated attack can find adversarial examples surrounded by decision boundaries that more accurately mimic the boundaries around benign examples.

To summarize, our contributions are:

1. We demonstrate OPTMARGIN, a new attack that evades region classification systems with low-distortion adversarial examples.

2. We introduce an analysis of decision boundaries around an input instance that explains the effectiveness of OPTMARGIN adversarial examples and also shows the attack's weaknesses.

3. We demonstrate the expressiveness of decision boundary information by using it to classify different kinds of input instances.

We have released the code we used at `https://github.com/sunblaze-ucb/decision-boundaries`.

## 2 BACKGROUND AND EXPERIMENTAL SETUP

In this paper, we study adversarial examples on the task of image classification. In image classification, a model $f$ takes an image $x \in \mathbb{R}^{\text{height} \times \text{width} \times \text{channels}}$ and assigns it a label $f(x) \in C$ from a set of classes $C$. These input instances come from a continuous high-dimensional space, while the output is discrete.

### 2.1 DATASETS

We use two popular academic image classification datasets for our experiments: MNIST, consisting of black-and-white handwritten digits (LeCun, 1998), and CIFAR-10, consisting of small color pictures (Krizhevsky & Hinton, 2009). In MNIST, the images' pixel values are in the range $[0, 1]$; in CIFAR-10, they are in $[0, 255]$. Additionally, we report similar experimenal results on a small subset of ImageNet in Appendix D.

### 2.2 ADVERSARIAL EXAMPLES

Adversarial examples are slightly perturbed versions of correctly classified input instances, which are misclassified. Attacks that generate adversarial examples can be *targeted*, producing examples that are incorrectly classified as an attacker-chosen class, or *untargeted*, producing examples that are misclassified as any class other than the correct one. For simplicity, we focus our analysis on untargeted attacks.

The amount of perturbation used to generate an adversarial example from the original input instance is called the example's *distortion*. In this paper, we quantify the distortion using the root-mean-square (RMS) distance metric between the original input instance and the adversarial example.

### 2.3 DEFENSES

Research on defenses against adversarial examples has explored many different techniques, both for detecting and correcting adversarial examples. In this paper, we discuss two recent defenses (from among many): adversarial training with examples generated by projected gradient descent (PGD) (Madry et al., 2017) and region classification (Cao & Gong, 2017).

Adversarial training modifies the training procedure, substituting a portion of the training examples (all of them, in the case of Madry et al.) with adversarial examples. Madry et al. perform adversarial

training using PGD, an attack that follows the gradient of the model's loss function for multiple steps to generate an adversarial example.

We give an overview of region classification in Section 3.1.

## 2.4 MODELS

In this paper, for each dataset, we perform experiments on two models trained from one architecture. For MNIST, the architecture is a convolutional neural network;[1] for CIFAR-10, a wide ResNet w32-10.[2] In order to study the effect of PGD adversarial training on a model's decision regions, from each dataset, we use a defended model trained with the PGD adversarial training defense and an undefended model trained with normal examples. The PGD adversarial training on MNIST used an $L_\infty$ perturbation limit of 0.3; on CIFAR-10, 8.

## 3 OPTMARGIN ATTACK ON REGION CLASSIFICATION

In this section, we develop a concrete example where limiting the analysis of a neighborhood to a small ball leads to evasion attacks on an adversarial example defense.

### 3.1 BACKGROUND: REGION CLASSIFICATION

Cao & Gong (2017) propose *region classification*, a defense against adversarial examples that takes the majority prediction on several slightly perturbed versions of an input, uniformly sampled from a hypercube around it. This approximates computing the majority prediction across the neighborhood around an input as a region. In contrast, the usual method of classifying only the input instance can be referred to as *point classification*.

Cao & Gong show that region classification approach successfully defends against low-distortion adversarial examples generated by existing attacks, and they suggest that adversarial examples robust to region classification, such as Carlini & Wagner's high-confidence attack, have higher distortion and can be detected by other means.

### 3.2 PROPOSED OPTMARGIN ATTACK

We introduce an attack, OPTMARGIN, which can generate low-distortion adversarial examples that are robust to small perturbations, like those used in region classification.

In our OPTMARGIN attack, we create a surrogate model of the region classifier, which classifies a smaller number of perturbed input points. This is equivalent to an ensemble of models $f_i(x) = f(x + v_i)$, where $f$ is the point classifier used in the region classifier and $v_i$ are perturbations applied to the input $x$. Our attack uses existing optimization attack techniques to generate an example that fools the entire ensemble while minimizing its distortion (Liu et al., 2017; He et al., 2017).

Let $Z(x)$ refer to the $|C|$-dimensional vector of class weights, in logits, that $f$ internally uses to classify image $x$. As in Carlini & Wagner's $L_2$ attack (2017b), we define a loss term for each model in our ensemble:

$$\ell_i(x') = \ell(x' + v_i) = \max\left(-\kappa,\ Z(x' + v_i)_y - \max\{Z(x' + v_i)_j : j \neq y\}\right)$$

This loss term increases when model $f_i$ predicts the correct class $y$ over the next most likely class. When the prediction is incorrect, the value bottoms out at $-\kappa$ logits, with $\kappa$ referred to as the confidence margin. In OPTMARGIN, we use $\kappa = 0$, meaning it is acceptable that the model just barely misclassifies its input. With these loss terms, we extend Carlini & Wagner's $L_2$ attack (2017b) to use an objective function that uses the sum of these terms. Whereas Carlini & Wagner would have one $\ell(x')$ in the minimization problem below, we have:

---

[1] https://github.com/MadryLab/mnist_challenge
[2] https://github.com/MadryLab/cifar10_challenge

$$\text{minimize} \quad ||x' - x||_2^2 + c \cdot (\ell_1(x') + ... + \ell_n(x'))$$

We use 20 classifiers in the attacker's ensemble, where we choose $v_1, ..., v_{19}$ to be random orthogonal vectors of uniform magnitude $\varepsilon$, and $v_{20} = 0$. This choice is meant to make it likely for a random perturbation to lie in the region between the $v_i$'s. Adding $f_{20}(x) = f(x)$ to the ensemble causes the attack to generate examples that are also adversarial under point classification.

For stability in optimization, we used fixed values of $v_i$ throughout the optimization of the attack. This technique was previously used in Carlini & Wagner's attack (2017a) on Feinman et al.'s stochastic dropout defense (2017).

## 3.3 DISTORTION EVALUATION

We compare the results of our OPTMARGIN attack with Carlini & Wagner's $L_2$ attack (2017b) with low confidence $\kappa = 0$, which we denote OPTBRITTLE, and with high confidence $\kappa = 40$, which we denote OPTSTRONG, as well as FGSM (Goodfellow et al., 2015) with $\epsilon = 0.3$ (in $L_\infty$ distance) for MNIST and 8 for CIFAR-10. In our OPTMARGIN attacks, we use $\varepsilon = 0.3$ (in RMS distance) for MNIST and $\varepsilon = 8$ for CIFAR-10. Figure 5 in the appendix shows a sample of images generated by each method. Table 1 shows the average distortion (amount of perturbation used) across a random sample of adversarial examples.

| Examples | MNIST | | | | CIFAR-10 | | | |
|---|---|---|---|---|---|---|---|---|
| | Normal | | Adv tr. | | Normal | | Adv tr. | |
| OPTBRITTLE | 100% | 0.0732 | 100% | 0.0879 | 100% | 0.824 | 100% | 3.83 |
| OPTMARGIN (ours) | 100% | 0.158 | 100% | 0.168 | 100% | 1.13 | 100% | 4.08 |
| OPTSTRONG | 100% | 0.214 | 28% | 0.391 | 100% | 2.86 | 73% | 37.4 |
| FGSM | 91% | 0.219 | 6% | 0.221 | 82% | 8.00 | 36% | 8.00 |

Table 1: Success rate (%) and average distortion (RMS) of adversarial examples generated by different attacks. On MNIST, the level of distortion in OPTMARGIN examples is visible to humans, but the original class is still distinctly visible (see Figure 5 in the appendix for sample images).

On average, the OPTMARGIN examples have higher distortion than OPTBRITTLE examples (which are easily corrected by region classification) but much lower distortion than OPTSTRONG examples.

The OPTSTRONG attack produces examples with higher distortion, which Cao & Gong discount; they suggest that these are easier to detect through other means. Additionally, the OPTSTRONG attack does not succeed in finding adversarial examples with a satisfactory confidence margins for all images on PGD adversarially trained models.[3] The FGSM samples are also less successful on the PGD adversarially trained models. The average distortion reported in Table 1 is averaged over only the successful adversarial examples in these two cases. The distortion and success rate can be improved by using intermediate confidence values, at the cost of lower robustness. Due to the low success rate and high distortion, we do not consider OPTSTRONG attacks in the rest of our experiments.

## 3.4 EVADING REGION CLASSIFICATION

We evaluate the effectiveness of our OPTMARGIN attack by testing the generated examples on Cao & Gong's region classification defense.

We use a region classifier that takes 100 samples from a hypercube around the input. Cao & Gong determined reasonable hypercube radii for similar models by increasing the radius until the region classifier's accuracy on benign data would fall below the accuracy of a point classifier. We use their reported values in our own experiments: 0.3 for a CNN MNIST classifier and 5.1 (0.02 of 255) for a ResNet CIFAR-10 classifier.

---

[3]We use the official implementation of Carlini & Wagner's high confidence attack, which does not output a lower-confidence adversarial example even if it encounters one.

In the following experiments, we test with a sample of 100 images from the test set of MNIST and CIFAR-10.

Table 2 shows the accuracy of four different configurations of defenses for each task: no defense (point classification with normal training), region classification (with normal training), PGD adversarial training (with point classification), and region classification with PGD adversarial training.

| | MNIST | | | | CIFAR-10 | | | |
| | Region cls. | | Point cls. | | Region cls. | | Point cls. | |
| Examples | Normal | Adv. tr. | Normal | Adv. tr. | Normal | Adv. tr. | Normal | Adv. tr. |
|---|---|---|---|---|---|---|---|---|
| Benign | 99% | 100% | 99% | 100% | 93% | 86% | 96% | 86% |
| FGSM | 16% | 54% | 9% | 94% | 16% | 55% | 17% | 55% |
| OPTBRITTLE | 95% | 89% | **0%** | **0%** | 71% | 79% | **0%** | **0%** |
| OPTMARGIN (ours) | **1%** | **10%** | **0%** | **0%** | **5%** | **5%** | **0%** | 6% |

Table 2: Accuracy of region classification and point classification on examples from different attacks. More effective attacks result in lower accuracy. The attacks that achieve the lowest accuracy for each configuration of defenses are shown in bold. We omit comparison with OPTSTRONG due to its disproportionately high distortion and low attack success rate.

Cao & Gong develop their own attacks against region classification, CW-$L_0$-A, CW-$L_2$-A, and CW-$L_\infty$-A. These start with Carlini & Wagner's low-confidence $L_0$, $L_2$, and $L_\infty$ attacks, respectively, and amplify the generated perturbation by some multiplicative factor. They evaluate these in a targeted attack setting. Their best result on MNIST is with CW-$L_2$-A with a $2\times$ amplification, resulting in 63% attack success rate. Their best result on CIFAR-10 is with CW-$L_\infty$-A with a $2.8\times$ amplification, resulting in 85% attack success rate. In our experiments with OPTMARGIN in an untargeted attack setting, we observe high attack success rates at similar increases in distortion.

These results show that our OPTMARGIN attack successfully evades region classification and point classification.

### 3.5 PERFORMANCE

Using multiple models in an ensemble increases the computational cost of optimizing adversarial examples, proportional to the number of models in the ensemble. Our optimization code, based on Carlini & Wagner's, uses 4 binary search steps with up to 1,000 optimization iterations each. In our slowest attack, on the PGD adversarially trained CIFAR-10 model, our attack takes around 8 minutes per image on a GeForce GTX 1080.

Although this is computationally expensive, an attacker can generate successful adversarial examples with a small ensemble (20 models) compared to the large number of samples used in region classification (100)—the slowdown factor is less for the attacker than for the defender.

## 4 DECISION BOUNDARIES AROUND ADVERSARIAL EXAMPLES

We have shown that examining a small ball around a given input instance may not adequately distinguish OPTMARGIN adversarial examples. In this section, we introduce a more comprehensive analysis of the neighborhood around an input instance. We study the *decision boundaries* of a model—the surfaces in the model's input space where the output prediction changes between classes. We examine benign and adversarial examples in terms of the decision boundaries surrounding them in the input space.

Specifically, we consider the distance to the nearest boundary in many directions (Section 4.1) and adjacent decision regions' classes (Section 4.2).

### 4.1 DECISION BOUNDARY DISTANCE

To gather information on the sizes and shapes of a model's decision regions, we estimate the distance to a decision boundary in a sample of random directions in the model's input space, starting from a

given input point. In each direction, we estimate the distance to a decision boundary by computing the model's prediction on perturbed inputs at points along the direction. In our experiments, we check every 0.02 units (in RMS distance) for MNIST and every 2 units for CIFAR-10. When the model's prediction on the perturbed image changes from the prediction on the original image (at the center), we use that distance as the estimate of how far the decision boundary is in that direction. When the search encounters a boundary this way, we also record the predicted class of the adjacent region.

For CIFAR-10, we perform this search over a set of 1,000 random orthogonal directions (for comparison, the input space is 3,072-dimensional). For MNIST, we search over 784 random orthogonal directions (the entire dimensionality of the input space) in both positive and negative directions, for a total of 1,568 directions.

### 4.1.1 INDIVIDUAL INSTANCES

Figure 1 shows the decision boundary distances for a typical set of a benign example and adversarial examples generated as described in Section 3 (OPTBRITTLE is an easily mitigated C&W low-confidence $L_2$ attack; OPTMARGIN is our method for generating robust examples; FGSM is the fast gradient sign method from Goodfellow et al. (2015)). It shows these attacks applied to models trained normally and models trained with PGD adversarial examples. See Figure 6 in the appendix for a copy of this data plotted in $L_\infty$ distance.

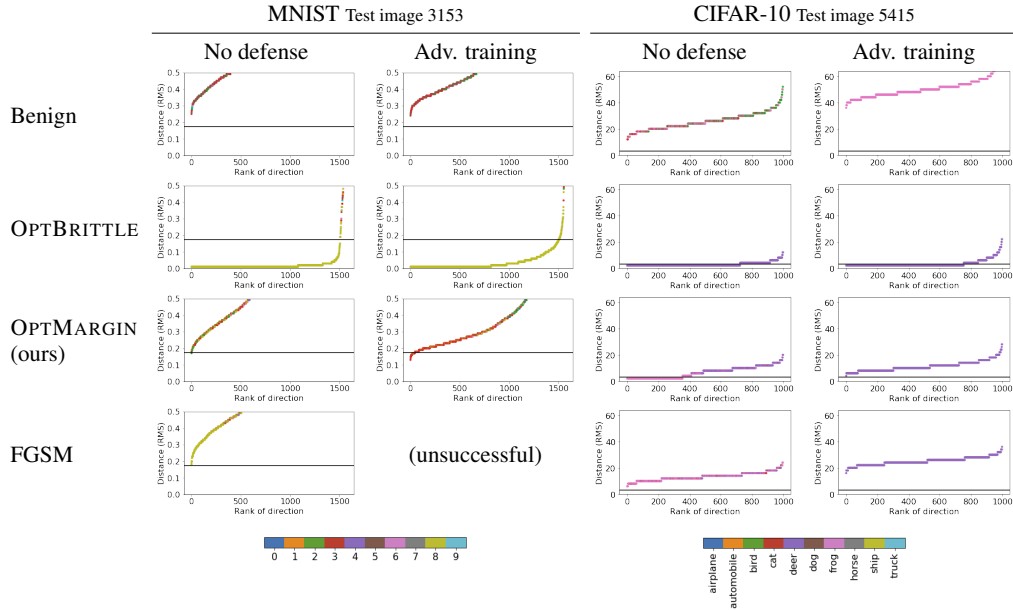

Figure 1: Decision boundary distances (RMS) from single sample images, plotted in ascending order. Colors represent the adjacent class to an encountered boundary. A black line is drawn at the expected distance of an image sampled during region classification. Results are shown for models with normal training and models with PGD adversarial training. For MNIST, original example correctly classified 8 (yellow); OPTBRITTLE and OPTMARGIN examples misclassified as 5 (brown); FGSM example misclassified as 2 (green). For CIFAR-10, original example correctly classified as DEER (purple); OPTBRITTLE, OPTMARGIN, and FGSM examples misclassified as HORSE (gray).

The boundary distance plots for examples generated by the basic optimization attack are strikingly different from those for benign examples. As one would expect from the optimization criteria, they are as close to the boundary adjacent to the original class as possible, in a majority of the directions. These plots depict why region classification works well on these examples: a small perturbation in nearly every direction crosses the boundary to the original class.

For our OPTMARGIN attack, the plots lie higher, indicating that the approach successfully creates a margin of robustness in many random directions. Additionally, in the MNIST examples, the original

class is not as prominent in the adjacent classes. Thus, these examples are challenging for region classification both due to robustness to perturbation and due to the neighboring incorrect decision regions.

### 4.1.2 SUMMARY STATISTICS.

We summarize the decision boundary distances of each image by looking at the minimum and median distances across the random directions. Figure 2 shows these representative distances for a sample of correctly classified benign examples and successful adversarial examples. See Figure 7 in the appendix for a copy of this data plotted in $L_\infty$ distance.

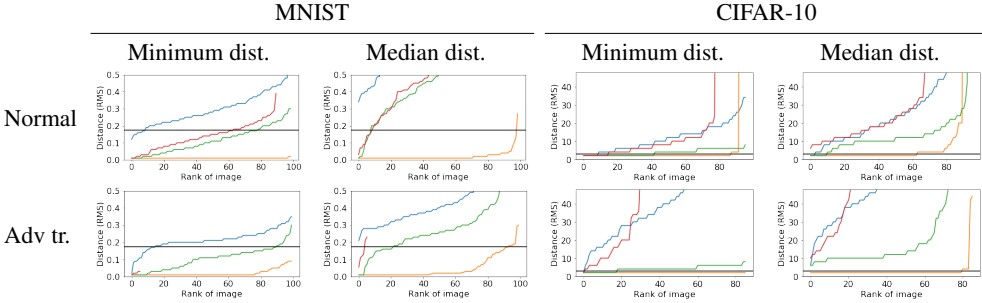

Figure 2: Minimum and median decision boundary distances across random directions, for a sample of images. **Blue**: Benign. **Red**: FGSM. **Green**: OPTMARGIN (ours). **Orange**: OPTBRITTLE. Each statistic is plotted in ascending order. A black line is drawn at the expected distance of images sampled by region classification.

These plots visualize why OPTMARGIN and FGSM examples, in aggregate, are more robust to random perturbations than the OPTBRITTLE attack. The black line, which represents the expected distance that region classification will check, lies below the green OPTMARGIN line in the median distance plots, indicating that region classification often samples points that match the adversarial example's incorrect class. OPTMARGIN and FGSM examples, however, are still less robust than benign examples to random noise.

Unfortunately, on MNIST, no simple threshold on any one of these statistics accurately separates benign examples (blue) from OPTMARGIN examples (green). At any candidate threshold (a horizontal line), there is either too much of the blue line below it (false positives) or too much of the green line above it (false negatives).

PGD adversarial training on the MNIST architecture results in decision boundaries closer to the benign examples, reducing the robustness to random perturbations. In CIFAR-10, however, the opposite is observed, with boundaries farther from benign examples in the PGD adversarially trained model. The effect of PGD adversarial training on the robustness of benign examples to random perturbations is not universally beneficial nor harmful.

### 4.2 ADJACENT CLASS PURITY

Another observation from plots like those in Figure 1 is that adversarial examples tend to have most directions lead to a boundary adjacent to a single class. We compute the *purity of the top k classes* around an input image as the largest cumulative fraction of random directions that encounter a boundary adjacent to one of $k$ classes.

Figure 3 shows the purity of the top $k$ classes averaged across different samples of images, for varying values of $k$. These purity scores are especially high for OPTBRITTLE adversarial examples compared to the benign examples. The difference is smaller in CIFAR-10, with the purity of benign examples being higher.

Region classification takes advantage of cases where the purity of the top 1 class is high, *and* the one class is the correct class, *and* random samples from the region are likely to be past those boundaries.

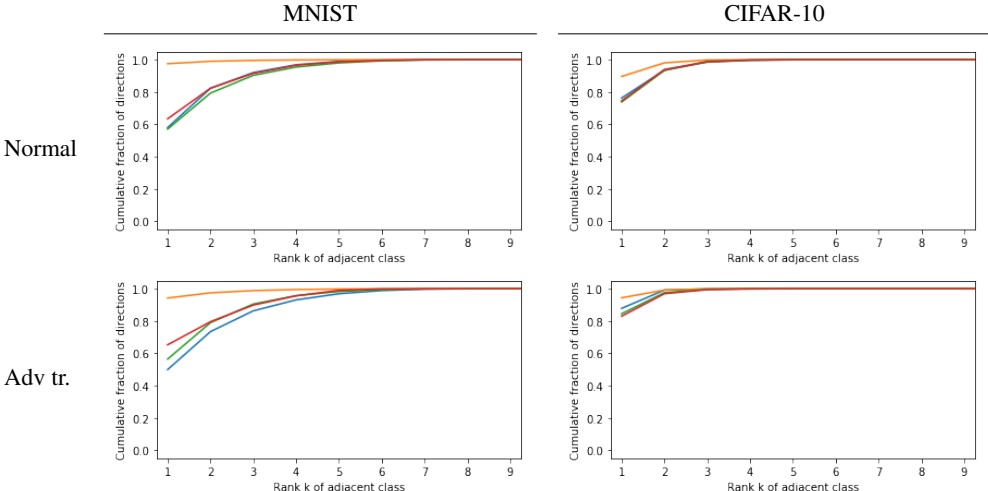

Figure 3: Average purity of adjacent classes around benign and adversarial examples.
**Orange**: OPTBRITTLE. **Red**: FGSM. **Green**: OPTMARGIN (ours). **Blue**: Benign. Curves that are lower on the left indicate images surrounded by decision regions of multiple classes. Curves that near the top at rank 1 indicate images surrounded almost entirely by a single class.

Adversarial examples generated by OPTMARGIN and FGSM are much harder to distinguish from benign examples in this metric.

## 5   DECISION BOUNDARY CLASSIFICATION

Cao & Gong's region classification defense is limited in its consideration of a hypercube region of a fixed radius, the same in all directions. We successfully bypassed this defense with our OPT-MARGIN attack, which created adversarial examples that were robust to small perturbations in many directions. However, the surrounding decision boundaries of these adversarial examples and benign examples are still different, in ways that sampling a hypercube would not reveal.

In this section, we propose a more general system for utilizing the neighborhood of an input to determine whether the input is adversarial. Our design considers the distribution of distances to a decision boundary in a set of randomly chosen directions and the distribution of adjacent classes—much more information than Cao & Gong's approach.

### 5.1   DESIGN

We ask the following question: Can information about the decision boundaries around an input be used to differentiate the adversarial examples generated using the current attack methods and benign examples? These adversarial examples are surrounded by distinctive boundaries on some models, such as the the PGD adversarially trained CIFAR-10 model (seen in Figure 2). However, this is not the case for either MNIST model, where no simple threshold can accurately differentiate OPTMARGIN adversarial examples from benign examples. In order to support both models, we design a classifier that uses comprehensive boundary information from many random directions.

We construct a neural network to classify decision boundary information, which we show in Figure 4. The network processes the distribution of boundary distances by applying two 1-D convolutional layers to a sorted array of distances. Then, it flattens the result, appends the first three purity scores, and applies two fully connected layers, resulting in a binary classification. We use rectified linear units for activation in internal layers. During training, we use dropout (Hinton et al., 2012) with probability 0.5 in internal layers.

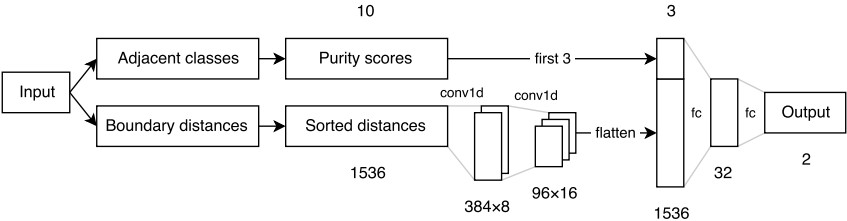

Figure 4: Architecture of our decision boundary classifier. Sizes are shown for our MNIST experiments.

## 5.2 EXPERIMENTAL RESULTS

We train with an Adam optimizer with a batch size of 128 and a learning rate of 0.001. For MNIST, we train on 8,000 examples (each *example* here contains both a benign image and an adversarial image) for 32 epochs, and we test on 2,000 other examples. For CIFAR-10, we train on 350 examples for 1,462 epochs, and we test on 100 other examples. We filtered these sets only to train on correctly classified benign examples and successful adversarial examples.

Table 3 shows the false positive and false negative rates of the model when using the hard max of the output. We had fewer successful adversarial examples from the FGSM attacks than for OPTBRITTLE and OPTMARGIN. We discuss the results of the corresponding decision boundary classification experiment on FGSM examples in Appendix C.

| | False pos. | False neg. | | Accuracy | |
| Training attack | Benign | OPTBRITTLE | OPTMARGIN | Our approach | Cao & Gong |
| --- | --- | --- | --- | --- | --- |
| | MNIST, normal training | | | | |
| OPTBRITTLE | 1.0% | 1.0% | 74.1% | | |
| OPTMARGIN | **9.6%** | 0.6% | 7.2% | 90.4% | 10% |
| | MNIST, PGD adversarial training | | | | |
| OPTBRITTLE | 2.6% | 2.0% | 39.8% | | |
| OPTMARGIN | 10.3% | 0.4% | 14.5% | | |
| | CIFAR-10, normal training | | | | |
| OPTBRITTLE | 5.3% | 3.2% | 56.8% | | |
| OPTMARGIN | 8.4% | 7.4% | 5.3% | 96.4% | 5% |
| | CIFAR-10, PGD adversarial training | | | | |
| OPTBRITTLE | 0.0% | 2.4% | 51.8% | | |
| OPTMARGIN | **3.6%** | 0.0% | 1.2% | | |

Table 3: False positive and false negative rates for the decision boundary classifier, trained on examples from one attack and evaluated examples generated by the same or a different attack. We consider the accuracy under the worst-case benign/adversarial data split (all-benign if false positive rate is higher; all-adversarial if false negative rate is higher), and we select the best choice of base model and training set. These best-of-worst-case numbers are shown in bold and compared with Cao & Gong's approach from Table 2.

This classifier achieves high accuracy on the attacks we study in this paper. These results suggest that our current best attack, OPTMARGIN, does not accurately mimic the distribution of decision boundary distances and adjacent classes. On MNIST, the model with normal training had better accuracy, while the model with PGD adversarial training had better accuracy on CIFAR-10. We do not have a conclusive explanation for this, but we do note that these were the models with decision boundaries being farther from benign examples (Figure 2). It remains an open question, however, whether adversaries can adapt their attacks to generate examples with surrounding decision boundaries that more closely match benign data.

## 5.3 PERFORMANCE

Assuming one already has a base model for classifying input data, the performance characteristics of this experiment are dominated by two parts: (i) collecting decision boundary information around given inputs and (ii) training a model for classifying the decision boundary information.

Our iterative approach to part (i) is expensive, involving many forward invocations of the base model. In our slowest experiment, with benign images on the PGD adversarially trained wide ResNet w32-10 CIFAR-10 model, it took around 70 seconds per image to compute decision boundary information for 1,000 directions on a GeForce GTX 1080. This time varies from image to image because our algorithm stops searching in a direction when it encounters a boundary. Collecting decision boundary information for OPTBRITTLE examples was much faster, for instance. Collecting information in fewer directions can save time, and should perform well as long as the samples adequately capture the distribution of distances and adjacent classes.

Part (ii) depends only on the number of directions, and the performance is independent of the base model's complexity. In our experiments, this training phase took about 1 minute for each model and training set configuration.

Running the decision boundary classifier on the decision boundary information is fast compared to the training and boundary collection.

## 6 CONCLUSION

We considered the benefits of examining large neighborhoods around a given input in input space. We demonstrated an effective OPTMARGIN attack against a region classification defense, which only considered a small ball of the input space around a given instance. We analyzed the neighborhood of examples generated by this new attack by looking at the decision boundaries around them, as well as the boundaries around benign examples and less robust adversarial examples. This analysis incorporated information from many directions in input space and from longer distances than previous work. We found that the comprehensive information about surrounding decision boundaries reveals there are still differences between our robust adversarial examples and benign examples. It remains to be seen how attackers might generate adversarial examples that better mimic benign examples' surrounding decision boundaries.

### ACKNOWLEDGMENTS

We thank Neil Gong for discussing his work with us, and we thank our anonymous reviewers for their helpful suggestions. This work was supported in part by Berkeley Deep Drive, the Center for Long-Term Cybersecurity, and FORCES (Foundations Of Resilient CybEr-Physical Systems), which receives support from the National Science Foundation (NSF award numbers CNS-1238959, CNS-1238962, CNS-1239054, CNS-1239166). Any opinions, findings, and conclusions or recommendations expressed in this material are those of the authors and do not necessarily reflect the views of the National Science Foundation.

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

## A    SAMPLE IMAGES

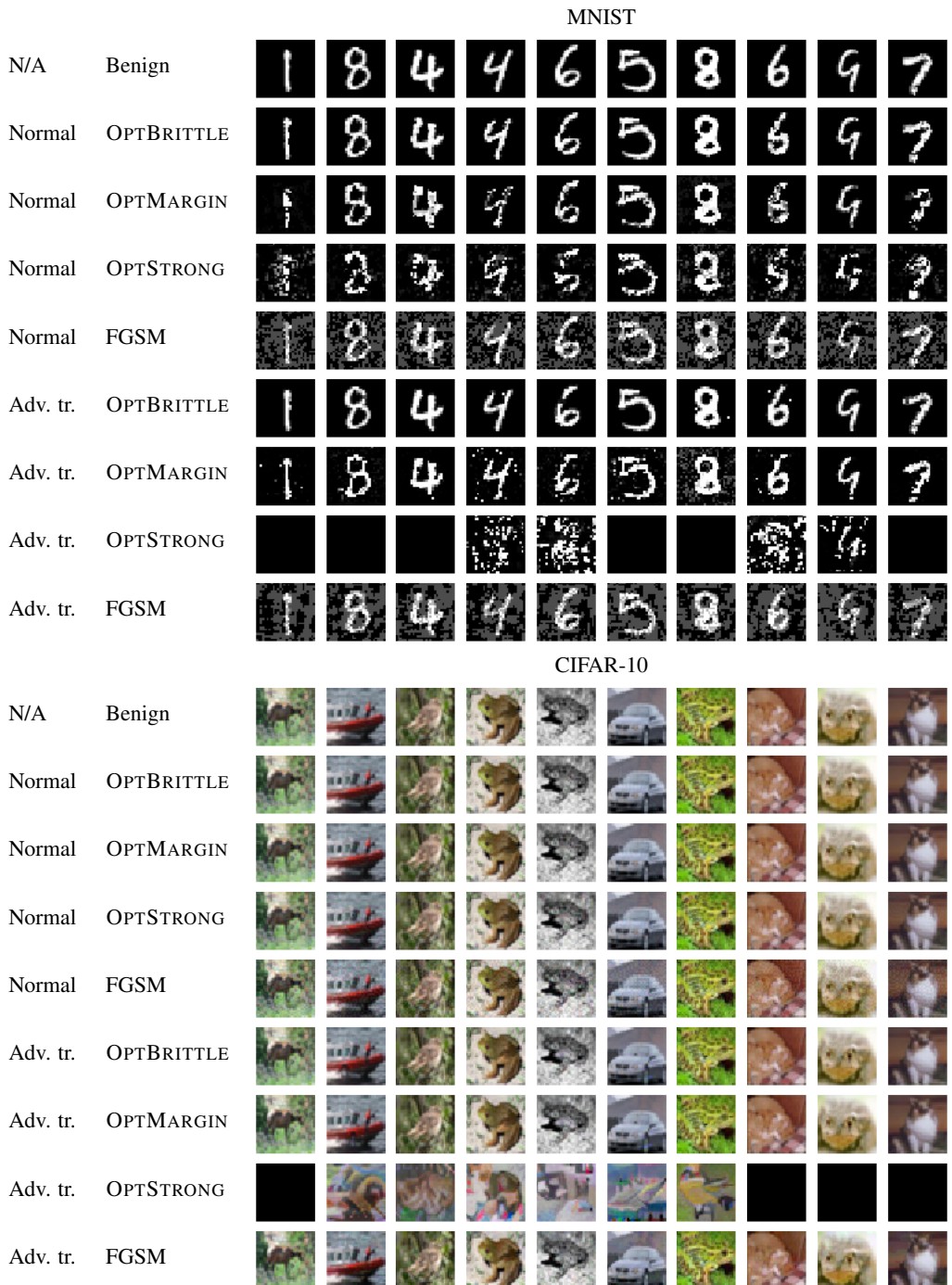

Figure 5: Adversarially perturbed images generated by different attack methods, for differently trained models, and their corresponding original images. Instances where the attack does not produce an example are shown as black squares.

## B   BOUNDARY DISTANCES IN $L_\infty$

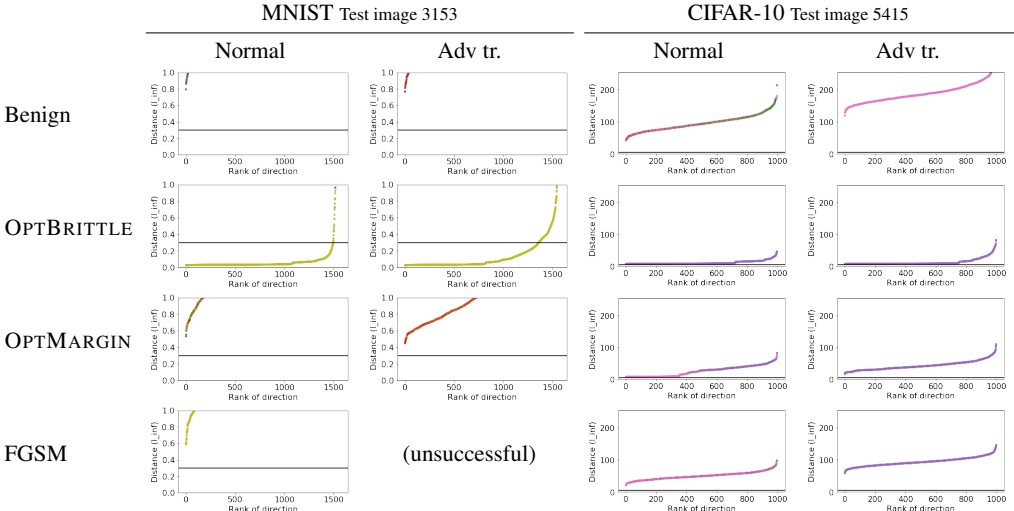

Figure 6: Equivalent of Figure 1, decision boundary distances from sample images, plotted in $L_\infty$ distance. A black line is drawn at the radius of the region used in region classification.

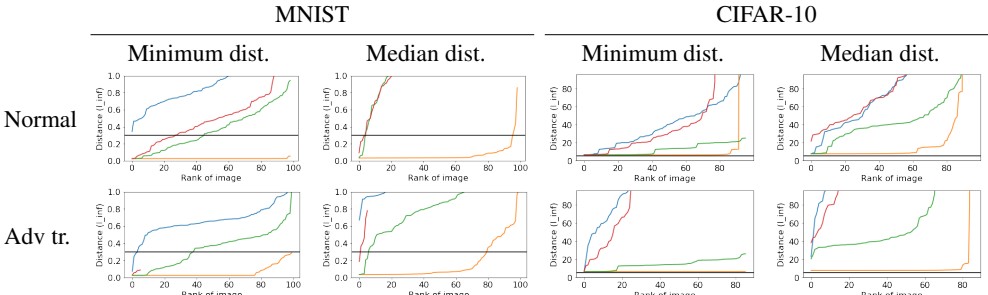

Figure 7: Equivalent of Figure 2, minimum and median decision boundary distances across random directions, plotted in $L_\infty$ distance. **Blue**: Benign. **Red**: FGSM. **Green**: OPTMARGIN (ours). **Orange**: OPTBRITTLE. A black line is drawn at the radius of the region used in region classification.

## C    CLASSIFYING FGSM DECISION BOUNDARIES

FGSM creates fewer successful adversarial examples, especially for adversarially trained models. The examples from our experiments ($\epsilon = 0.3$ for MNIST and 8 for CIFAR-10) have higher distortion than the OPTMARGIN examples and are farther away from decision boundaries. We trained a classifier on successful FGSM adversarial examples for normal models (without adversarial training). Table 4 shows the accuracy of these classifiers. PGD adversarial training is effective enough that we did not have many successful adversarial examples to train the classifier.

| Dataset | Normal training | |
| | False pos. | False neg. |
|---|---|---|
| MNIST | 7.0% | 12.8% |
| CIFAR-10 | 20.0% | 32.9% |

Table 4: False positive and false negative rates for the decision boundary classifier, trained and evaluated on FGSM examples.

## D    EXPERIMENTS ON IMAGENET

We perform a similar series of experiments on a small subset of ImageNet (Russakovsky et al., 2015), using Szegedy et al.'s Inception-ResNet model[4] (2017) in a top-1 classification task. We experiment with a small sample of 450 images from the validation set. We use a hypercube with radius 0.02 for region classification (the same relative size as for CIFAR-10, but for pixel values in the range $[0, 1]$), $\varepsilon = 8/255$ for OPTMARGIN (0.031), and $\epsilon = 8/255$ for FGSM. In experiments where we train and test a classifier, we divide the set into 350 images for training and 100 images for testing. These experiments use the same number of examples as our CIFAR-10 experiments, but relative to the scope of ImageNet, there are fewer than are needed to exercise all 1,000 classes in the dataset. Thus, the results in this section are more preliminary.

Table 5 summarizes the effectiveness of OPTMARGIN and other attacks on Cao & Gong's region classification defense and the effectiveness of decision boundary classification. The results are consistent with our experiments on MNIST and CIFAR-10, with OPTMARGIN having the highest attack success rate under region classification. However, our decision boundary classifier network accurately classifies OPTBRITTLE and OPTMARGIN adversarial examples. FGSM examples have much higher distortion and are less successful but are less accurately classified.

| Attack | Distortion | Top-1 accuracy | | Boundary classification | |
| | | Point cls. | Region cls. | False pos. | False neg. |
|---|---|---|---|---|---|
| Benign | N/A | 65% | 66% | N/A | N/A |
| OPTBRITTLE | 0.000 526 | **0%** | 64% | 1% | 0% |
| OPTMARGIN | 0.001 01 | **0%** | **16%** | 4% | 3% |
| FGSM | 0.0308 | 22% | 22% | 39% | 41% |

Table 5: Effectiveness of attacks on ImageNet. Reported in this table: average distortion (RMS) of successful examples, top-1 accuracy under point classification and region classification, and false positive and false negative rates of a decision boundary classifier trained on examples of the same attack.

Figure 8 shows images and surrounding decision boundaries for a sample validation image and adversarial examples created from it. Figure 9 presents summary statistics of decision boundary distance and the average purity of adjacent classes around a sample of validation images. Both the individual examples and summary statistics show that OPTBRITTLE examples are susceptible to classification changes under slight random perturbations, and OPTMARGIN examples are robust enough to withstand the random perturbations used in region classification. On ImageNet, FGSM

---

[4]https://github.com/tensorflow/models/tree/master/research/slim

examples show much higher robustness to random perturbations and more a composition of adjacent decision region classes more similar to benign images.

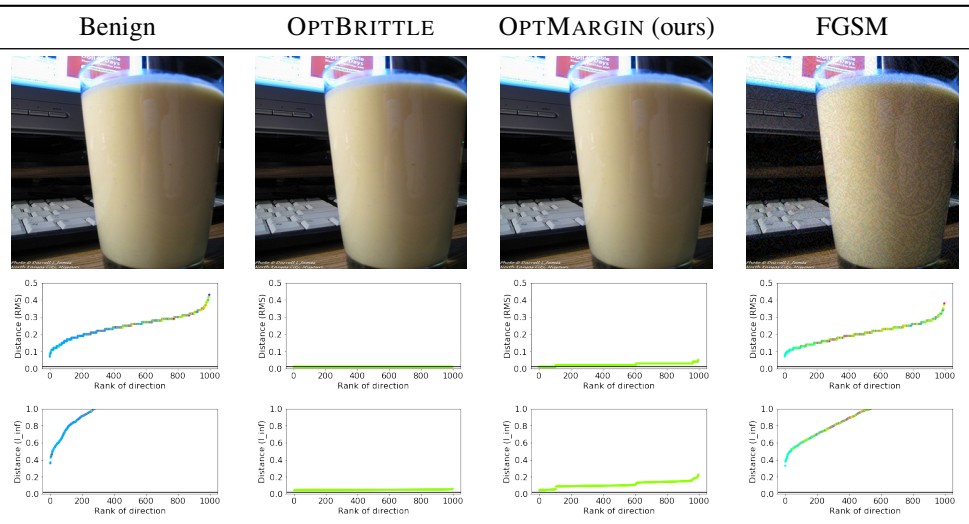

Figure 8: Images (top) and decision boundary distances in RMS distance (middle) and $L_\infty$ distance (bottom) based of a validation example from ImageNet and adversarial examples. Original example correctly classified as EGGNOG (lime); OPTBRITTLE example misclassified as ESPRESSO (cyan); OPTMARGIN example misclassified as BEER GLASS (pink); and FGSM example misclassified as COFFEE MUG (blue).

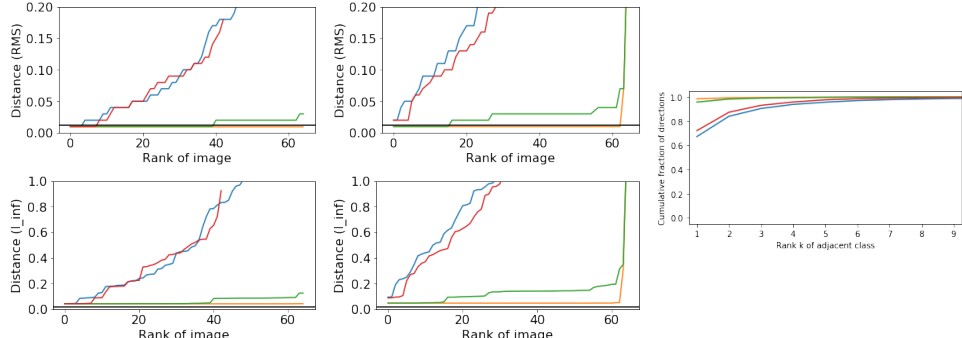

Figure 9: Minimum (left) and median (middle) decision boundary distances and average purity of adjacent classes (right) around a sample of ImageNet validation images. **Blue**: Benign. **Red**: FGSM. **Green**: OPTMARGIN (ours). **Orange**: OPTBRITTLE. In distance plots, a black line is drawn at the expected distance of images sampled by region classification in RMS distance plots (top) and radius of region in $L_\infty$ distance (bottom).

