# OpenReview forum: "Decision Boundary Analysis of Adversarial Examples"
_ICLR.cc/2018/Conference — Accept (Poster)_

### Official Review · AnonReviewer3 · 2017-11-25
**Review for "Decision Boundary Analysis of Adversarial Examples"**

**Rating:** 6
**Confidence:** 3

**Review:**

Summary of paper:

The authors present a novel attack for generating adversarial examples, deemed OptMargin, in which the authors attack an ensemble of classifiers created by classifying at random L2 small perturbations. They compare this optimization method with two baselines in MNIST and CIFAR, and provide an analysis of the decision boundaries by their adversarial examples, the baselines and non-altered examples.

Review summary:

I think this paper is interesting. The novelty of the attack is a bit dim, since it seems it's just the straightforward attack against the region cls defense. The authors fail to include the most standard baseline attack, namely FSGM. The authors also miss the most standard defense, training with adversarial examples. As well, the considered attacks are in L2 norm, and the distortion is measured in L2, while the defenses measure distortion in L_\infty (see detailed comments for the significance of this if considering white-box defenses). The provided analysis is insightful, though the authors mostly fail to explain how this analysis could provide further work with means to create new defenses or attacks.

If the authors add FSGM to the batch of experiments (especially section 4.1) and address some of the objections I will consider updating my score.

A more detailed review follows.


Detailed comments:

- I think the novelty of the attack is not very strong. The authors essentially develop an attack targeted to the region cls defense. Designing an attack for a specific defense is very well established in the literature, and the fact that the attack fools this specific defense is not surprising.

- I think the authors should make a claim on whether their proposed attack works only for defenses that are agnostic to the attack (such as PGD or region based), or for defenses that know this is a likely attack (see the following comment as well). If the authors want to make the second claim, training the network with adversarial examples coming from OptMargin is missing.

- The attacks are all based in L2, in the sense that the look for they measure perturbation in an L2 sense (as the paper evaluation does), while the defenses are all L_\infty based (since the region classifier method samples from a hypercube, and PGD uses an L_\infty perturbation limit). This is very problematic if the authors want to make claims about their attack being effective under defenses that know OptMargin is a possible attack.

- The simplest most standard baseline of all (FSGM) is missing. This is important to compare properly with previous work.

- The fact that the attack OptMargin is based in L2 perturbations makes it very susceptible to a defense that backprops through the attack. This and / or the defense of training to adversarial examples is an important experiment to assessing the limitations of the attack.

- I think the authors rush to conclude that "a small ball around a given input distance can be misleading". Wether balls are in L2 or L_\infty, or another norm makes a big difference in defense and attacks, given that they are only equivalent to a multiplicative factor of sqrt(d) where d is the dimension of the space, and we are dealing with very high dimensional problems. I find the analysis made by the authors to be very simplistic.

- The analysis of section 4.1 is interesting, it was insightful and to the best of my knowledge novel. Again I would ask the authors to make these plots for FSGM. Since FSGM is known to be robust to small random perturbations, I would be surprised that for a majority of random directions, the adversarial examples are brought back to the original class.

- I think a bit more analysis is needed in section 4.2. Do the authors think that this distinguishability can lead to a defense that uses these statistics? If so, how?

- I think the analysis of section 5 is fairly trivial. Distinguishability in high dimensions is an easy problem (as any GAN experiment confirms, see for example Arjovsky & Bottou, ICLR 2017), so it's not surprising or particularly insightful that one can train a classifier to easily recognize the boundaries.

- Will the authors release code to reproduce all their experiments and methods?

Minor comments:
- The justification of why OptStrong is missing from Table2 (last three sentences of 3.3) should be summarized in the caption of table 2 (even just pointing to the text), otherwise a first reader will mistake this for the omission of a baseline.

- I think it's important to state in table 1 what is the amount of distortion noticeable by a human.

=========================================

After the rebuttal I've updated my score, due to the addition of FSGM added as a baseline and a few clarifications. I now understand more the claims of the paper, and their experiments towards them. I still think the novelty, significance of the claims and protocol are still perhaps borderline for publication (though I'm leaning towards acceptance), but I don't have a really high amount of experience in the field of adversarial examples in order to make my review with high confidence.

---

> ### Author Response · Authors · 2018-01-05
> **Authors' response**
>
> We thank the reviewer for the helpful suggestions and comments.
> [Comparison with FGSM] In our updated version, we’ve added the corresponding experiments with FGSM adversarial examples throughout the paper. Thanks for the suggestion. In summary, FGSM was able to create some robust adversarial examples, but it also had higher distortion and lower success rate, especially on adversarially trained models.
> [Standard defense, adversarial training] The adversarially trained model we used in this paper (with PGD adversarial training) is already intended to defend against gradient-based attacks, such as OptMargin and the other attacks we experiment with in this paper. We do not refute Madry et al.’s claim that PGD adversarial training is effective against attacks bounded in L_inf distortion (2017), although we do not find that threat model to be realistic.
> [Attacks in L2, defenses in L_infinity] The discrepancy between the attacks’ focus on L_2 distance and the defenses’ focus on L_inf distance is definitely not the most elegant thing in this paper. However, our analysis of random orthogonal directions is simplest in a distance metric where all directions are uniform. Cao & Gong’s defense, in particular, was not specialized for L_inf-bounded attacks, and their paper evaluated it successfully against previous L_0- and L_2 attacks. Nevertheless, we find it interesting that an adversarial example which satisfies points sampled from a hypersphere would be generally robust enough against a defense that checks in a hypercube.
> > Designing an attack for a specific defense is very well established in the literature, and the fact that the attack fools this specific defense is not surprising.
> The choice of defenses and attacks used in this paper are meant to cover a variety of scenarios for looking at decision boundaries. The proposal of OptMargin as an attack intends, in part, to demonstrate an adaptive attack, but primarily to create images farther from decision boundaries. The result that OptMargin bypasses region classification at all, we think is interesting because (i) research in nondeterministic defenses has picked up recently, and it is expected to have an advantage of unpredictability even in white-box settings; and (ii) it succeeds with less distortion than previously known methods, including OptStrong and Cao & Gong’s CW-L_*-A.
> > If the authors want to make the second claim [that OptMargin would work against other, specialiazed defenses], training the network with adversarial examples coming from OptMargin is missing.
> We do not aim to make that claim. We claim that OptMargin creates adversarial examples that are robust to small random perturbations and that have high attack success rate against region classification, a specific defense that examines the decision regions around the input.
> > I think the authors rush to conclude that "a small ball around a given input distance can be misleading". Wether balls are in L2 or L_\infty, or another norm makes a big difference ...
> Thanks for bringing this up—we intentionally evaluate classifier boundaries in terms of both metric, and we see evidence for this in our decision boundary distance plots, both for the hypersphere (Figure 2, formerly Table 4) and the hypercube (Figure 7, formerly Table 9). Additionally, for the hypercube, we experimentally validate that the ball is consistent enough to fool a region classifier.
> > Do the authors think that this distinguishability can lead to a defense that uses these statistics?
> We’re not sure whether these, or even the mechanisms of Section 5 would be a good enough defense. The summary statistics in Sections 4.1.2 and 4.2 are not separable with good accuracy in some settings. There is a fraction of high-distortion FGSM attacks that can fool a classifier and have a convincing distribution of surrounding decision boundaries. We have yet to find out if an attacker can achieve this with a high success rate and, ideally, lower distortion.
> > Distinguishability in high dimensions is an easy problem
> Yes, it was our intention to use more data to make the problem easier. We agree, though, that once we have the decision boundary data and see how different it is across different kinds of examples, it is not as big of a leap to run it through a classifier and to expect it to work. But we definitely wanted to have an application to showcase how such information can be used and improve over previous techniques.
> > release code
> Yes, we intend to release our code, as well as the random values we used in our experiments upon acceptance.
> Thanks for you minor comments as well. We have made changes in our updated draft to address them. We have updated the caption of Table 2 regarding the omission of OptStrong, and we have added an explanation to Table 1 about the visibility of perturbations.

---

### Official Review · AnonReviewer2 · 2017-11-28
**Extensive experiments and reasonable analysis of the results**

**Rating:** 6
**Confidence:** 2

**Review:**

Compared to previous studies, this paper mainly claims that the information from larger neighborhoods (more directions or larger distances) will better characterize the relationship between adversarial examples and the DNN model.

The idea of employing ensemble of classifiers is smart and effective. I am curious about the efficiency of the method.

The experimental study is extensive. Results are well discussed with reasonable observations. In addition to examining the effectiveness, authors also performed experiments to explain why OPTMARGIN is superior. Authors are suggested to involve more datasets to validate the effectiveness of the proposed method.

Table 5 is not very clear. Authors are suggested to discuss in more detail.

---

> ### Author Response · Authors · 2018-01-05
> **Authors' response**
>
> Thanks for reviewing our paper.
>
> In our updated draft, we’ve added some small-scale experiments on ImageNet data, in Appendix D. Thanks for the suggestion. The OptMargin attack is effective against region classification in these new experiments too.
>
> We’ve added some interpretation guidelines to the caption of Table 5 (the adjacent class purity plots, changed to Figure 3 in the updated version).

---

### Official Review · AnonReviewer1 · 2017-12-01
**Method that successfully attacks other existing defense methods, and present a method that can successfully defend this attack**

**Rating:** 6
**Confidence:** 3

**Review:**

The paper presents a new approach to generate adversarial attacks to a neural network, and subsequently present a method to defend a neural network from those attacks. I am not familiar with other adversarial attack strategies aside from the ones mentioned in this paper, and therefore I cannot properly assess how innovative the method is.

My comments are the following:

1- I would like to know if benign examples are just regular examples or some short of simple way of computing adversarial attacks.

2- I think the authors should provide a more detailed and formal description of the OPTMARGIN method. In section 3.2 they explain that "Our attack uses existing optimization attack techniques to...", but one should be able to understand the method without reading further references. Specially a formal representation of the method should be included.

3- Authors mention that OPTSTRONG attack does not succeed in finding adversarial examples ("it succeeds on 28% of the samples on MNIST;73% on CIFAR-10"). What is the meaning of success rate in here? Is it the % of times that the classifier is confused?

4- OPTSTRONG produces images that are notably more distorted than OPTBRITTLE (by RMS and also visually in the case of MNIST). So I actually cannot tell which method is better, at least in the MNIST experiment. One could do a method that completely distort the image and therefore will be classified with as a class. But adversarial images should be visually undistinguishable from original images. Generated CIFAR images seem similar than the originals, although CIFAR images are very low resolution, so judging this is hard.

4- As a side note, it would be interesting to have an explanation about why region classification is providing a worse accuracy than point classification for CIFAR-10 benign samples.

As a summary, the authors presented a method that successfully attacks other existing defense methods, and present a method that can successfully defend this attack. I would like to see more formal definitions of the methods presented. Also, just by looking at RMS it is expected that this method works better than OPTBRITTLE, since the images are more distorted. It would be needed to have a way of visually evaluate the similarity between original images and generated images.

---

> ### Author Response · Authors · 2018-01-05
> **Authors' response**
>
> Thanks for the comments and questions. Here are our responses and the corresponding changes we’ve made in our updated draft.
>
> 1. (What are benign examples?) The benign examples are just taken directly from the test set.
>
> 2. (More detail on how OptMargin works) Agreed. We’ve added an overview of the technique to Section 3.2, which should cover the relevant parts of the cited work.
>
> 3. (What is the success rate for OptStrong?) It’s the fraction of cases when the attack generates an image that’s misclassified *with high enough of a confidence margin* (40 logits in our experiments). Note that the official implementation of Carlini & Wagner’s high-confidence attack (referred to as OptStrong in in this paper) attack outputs a blank black image if the internal optimization procedure does not encounter a satisfactory image, even if it does encounter an image that would fool the classifier but with a lower confidence margin.
>
> 4. (OptStrong is heavily distorted) That’s a good point that an indistinguishable example would be better (more stealthy, lower cost by some measure, etc.). But the defender wouldn’t have an unmodified version of the image to compare against. The distortion numbers (Table 1) tell part of the story of how much the adversarial examples are changed. Internally, we like to look at sample images (Appendix A) and make sure the original class is still clearly visible to humans. Our opinion is that OptStrong images are less recognizable than OptMargin’s.
>
> 4. (Why is region classification worse on CIFAR-10?) Our intuition is that hypercubes, L_inf distances, and the like are especially well suited for MNIST, because the dataset is black and white. Random perturbations that change black to dark gray can be systematically ignored by a model that’s smart enough. CIFAR-10 deals with colors that use the whole gamut of pixel values, so it should be more sensitive to small changes bounded by an L_inf distance. Experimentally, we evaluated a model made with PGD adversarial training, which is trained not to be sensitive to these small perturbations, and the result is that the accuracy is lower than that of the model without adversarial training, but there’s no accuracy drop between point classification and region classification (Table 2).

---

### Author Response · Authors · 2018-01-07
**Summary of changes to the manuscript**

# Added experiments
- Expanded the experiments with the addition of FGSM as an attack method, as a well known baseline. (throughout)
- Repeated our experiments on a small subset of ImageNet, as an example of a more realistic dataset. (Appendix D)

# Writing changes
- Added a detailed description of how OptMargin works (Section 3.2).
- Added a note to the distortion levels (Table 1) to describe them qualitatively.
- Clarified why OptStrong (a high-confidence Carlini & Wagner attack) sometimes does not output a perturbed image (Section 3.3).
- Added a note to the accuracy of point- and region classification under attack (Table 2) to recap why OptStrong is omitted from comparison.
- Reduced a broader claim that “examining a small ball around a given input instance can be misleading ...” to be specific to evidence: “examining ... may not adequately distinguish OptMargin adversarial examples,” (Section 4).
- Added interpretation guidelines to the average purity of adjacent classes plots (Figure 3, formerly Table 5).

# Other changes
- Added links to the classification models we used. (Section 2.4)
- Fixed some grammatical errors.
- Added color to legend-like information in figure captions.
- Improved the styling of tables.
- Improved the parallelism in table headings, now “Normal” vs “Adv tr.,” previously “No defense” vs “PGD adv.”
- Changed collections of graphs to be Figures instead of Tables.

---

### Decision · Program_Chairs · 2018-01-29
**ICLR 2018 Conference Acceptance Decision**

**Decision:**

Accept (Poster)

**Comment:**

Authors propose an approach to generation of adversarial examples that jointly examine the effects to classification within a local neighborhood, to yield a more robust example. This idea is taken a step further for defense, whereby the classification boundaries within a local neighborhood of a presented example are examined to determine if the data was adversarially generated or not.


Pro:
- The idea of examining local neighborhoods around data points appears new and interesting.
- Evaluation and investigation is thorough and insightful.
- Authors made reasonable attempts to address reviewer concerns.

Con
 - Generation of adversarial examples an incremental improvement over prior methods